# Predicting Mortality Using Machine Learning Algorithms in Patients Who Require Renal Replacement Therapy in the Critical Care Unit

**DOI:** 10.3390/jcm11185289

**Published:** 2022-09-08

**Authors:** Hsin-Hsiung Chang, Jung-Hsien Chiang, Chi-Shiang Wang, Ping-Fang Chiu, Khaled Abdel-Kader, Huiwen Chen, Edward D. Siew, Jonathan Yabes, Raghavan Murugan, Gilles Clermont, Paul M. Palevsky, Manisha Jhamb

**Affiliations:** 1Division of Nephrology, Department of Internal Medicine, Antai Medical Care Corporation Antai Tian-Sheng Memorial Hospital, Donggang 928, Taiwan; 2Department of Computer Science and Information Engineering, National Cheng Kung University, Tainan 701, Taiwan; 3Renal-Electrolyte Division, Department of Medicine, University of Pittsburgh, Pittsburgh, PA 15213, USA; 4Division of Nephrology, Department of Internal Medicine, Changhua Christian Hospital, Changhua 500, Taiwan; 5Department of Hospitality Management, MingDao University, Changhua 500, Taiwan; 6Division of Nephrology and Hypertension, Vanderbilt University Medical Center, Nashville, TN 37011, USA; 7Vanderbilt Center for Kidney Disease (VCKD) and Integrated Program for AKI Research (VIP-AKI), Nashville, TN 37011, USA; 8Tennessee Valley Health Systems (TVHS), Veteran’s Health Administration, Nashville, TN 37212, USA; 9Division of General Internal Medicine, Department of Medicine, University of Pittsburgh, Pittsburgh, PA 15213, USA; 10Program for Critical Care Nephrology, CRISMA, Department of Critical Care Medicine, University of Pittsburgh School of Medicine, Pittsburgh, PA 15213, USA; 11Department of Critical Care Medicine, University of Pittsburgh, Pittsburgh, PA 15213, USA; 12Kidney Medicine Section, VA Pittsburgh Healthcare System, Pittsburgh, PA 15240, USA

**Keywords:** machine learning algorithm, mortality, acute kidney injury, renal replacement therapy

## Abstract

Background: General severity of illness scores are not well calibrated to predict mortality among patients receiving renal replacement therapy (RRT) for acute kidney injury (AKI). We developed machine learning models to make mortality prediction and compared their performance to that of the Sequential Organ Failure Assessment (SOFA) and HEpatic failure, LactatE, NorepInephrine, medical Condition, and Creatinine (HELENICC) scores. Methods: We extracted routinely collected clinical data for AKI patients requiring RRT in the MIMIC and eICU databases. The development models were trained in 80% of the pooled dataset and tested in the rest of the pooled dataset. We compared the area under the receiver operating characteristic curves (AUCs) of four machine learning models (multilayer perceptron [MLP], logistic regression, XGBoost, and random forest [RF]) to that of the SOFA, nonrenal SOFA, and HELENICC scores and assessed calibration, sensitivity, specificity, positive (PPV) and negative (NPV) predicted values, and accuracy. Results: The mortality AUC of machine learning models was highest for XGBoost (0.823; 95% confidence interval [CI], 0.791–0.854) in the testing dataset, and it had the highest accuracy (0.758). The XGBoost model showed no evidence of lack of fit with the Hosmer–Lemeshow test (*p* > 0.05). Conclusion: XGBoost provided the highest performance of mortality prediction for patients with AKI requiring RRT compared with previous scoring systems.

## 1. Introduction

Acute kidney injury (AKI) is a common and significant problem in intensive care units (ICU), with incidence rates reportedly as high as 50% of patients admitted [1]. Up to 25% of AKI patients in the ICU require renal replacement therapy (RRT) [1,2]. Despite advancements in the performance and technology of RRT, the mortality rate of those patients remains 30% to 50% [3,4]. Although the outcomes of these patients are likely related partly to the severity of their underlying diseases, having clinical tools that can accurately and reliably provide prognostic predictions is important to aid in clinical decision-making.

General severity of illness scores have been used to predict ICU mortality. For example, the Acute Physiology And Chronic Health Evaluation (APACHE) and Simplified Acute Physiology Score (SAPS) have been developed since the 1980s. They provide adequate prediction of in-hospital and ICU mortality of all ICU patients regardless of ICU type [5,6,7,8]. The Sequential Organ Failure Assessment (SOFA) score is also used for hospital and ICU mortality prediction [9]. However, studies have suggested that these traditional models are not reliable for the AKI populations who need RRT in ICU [10,11,12]. Instead, models using data at RRT initiation have performed better at mortality prediction. Some of these models have shown good performance for mortality prediction but have limited results during external validation [13,14,15,16].

Recently, machine learning models have been broadly applied from disease diagnosis to mortality prediction. They are expected to capture nonlinear interactions from high complexity data and consider all data points for continuous data, thus providing more accurate risk prediction than traditional models. We developed machine learning algorithms using data collected from the Medical Information Mart for Intensive Care (MIMIC-III) [17] and eICU Collaborative Research (eICU-CRD) databases [18] and compared the performance of the results to that of the SOFA [19]; nonrenal SOFA [20]; and HEpatic failure, LactatE, NorepInephrine, medical Condition, and Creatinine (HELENICC) [16] scores in 30-day mortality prediction for AKI patients requiring RRT.

## 2. Materials and Methods

### Data Sources

This retrospective observational cohort study was performed using two publicly available ICU datasets (MIMIC-III and eICU-CRD). MIMIC-III (53,423 ICU admissions between 2001 and 2012) was released by the Massachusetts Institute of Technology Laboratory for Computational Physiology (MIT-LCP) from a single tertiary care hospital (Beth Israel Deaconess Medical Center) in 2016 [17]. eICU-CRD (approximately 200,000 ICU admissions between 2014 and 2015) is a multicenter critical care database from rural/nonacademic hospitals across the United States made available by Philips Healthcare with the help of researchers from MIT-LCP in 2018 [18]. There is no overlap of patients included in these two databases [18].

We included adults ≥18 years old who received RRT (intermittent hemodialysis or continuous RRT [CRRT]) for AKI in the ICU. AKI was defined by the creatinine change level and diagnosis codes in this study. We only used creatinine criteria due to the unreliable urine data in the retrospective databases. Patients were included when they did not have at least 2 creatinine data but had AKI as a diagnosis using ICD-9 codes (Appendix A) or by maxium–miminum change of creatinine ≥ 0.3 mg/dL from ICU admission to RRT. If a patient had been admitted to the ICU multiple times during one hospitalization course, data from the first ICU admission were extracted for study. Patients with a history of end-stage kidney disease who underwent chronic peritoneal dialysis (PD) or hemodialysis (HD) were excluded from the study, as were those with chronic kidney disease (CKD) stages 4 and 5 based on ICD-9 codes (Appendix A), because advanced CKD patients who develop AKI are more likely to survive the episode of AKI [21]. Patients with a history of any organ transplant were also excluded, as they may have other confounding risk variables that affect mortality.

## 3. Model Development

### 3.1. Predictors

The variables of our models consisted of demographics, medical history, mechanical ventilation use, FiO_2_, vital signs, laboratory tests, and medications (diuretics, vasopressors). The mechanical ventilation, vital signs, lab tests, and medications were recorded within 24 h before RRT initiation. Relevant past medical history, extracted from database records using ICD-9 codes (Appendix A), included diabetes mellitus (DM), CKD, hypertension (HTN), congestive heart failure (CHF), liver cirrhosis (LC), and cancer. We used mean values of lab tests, FiO_2_, Glasgow Coma Scales (GCS), mean arterial pressure (MAP), and respiratory and heart rates (HR). FiO_2_ was from a laboratory test in MIMIC and from a respiratory chart in eICU.

For laboratory tests, we used mean values of all variables recorded within 24 h before the first dialysis therapy initiation date, because some laboratory data values would have been influenced by dialysis. Appendix A reveals the percentage of missing data in laboratory tests. We excluded variables with >30% missing values. Multiple imputation by chained equations (MICE) with five imputed datasets was used to derive the missing values of the laboratory tests and vital signs, and the results were pooled using the MICE package [22]. The missing values were completed, handled by MICE, and then the imputed data were used to build models.

We modified the codes found in the public domain at https://github.com/nus-mornin-lab/oxygenation_kc (accessed on 5 February 2020) and https://github.com/MIT-LCP/mimic-code/tree/master/concepts/severityscores (accessed on 5 February 2020) to calculate a SOFA score using variables collected within 24 h before RRT start in eICU and MIMIC based on methods in the original study [19]. We also calculated the nonrenal SOFA score, which was calculated by the total SOFA score minus the points for the renal system [20]. For patients with missing variables, SOFA and nonrenal SOFA scores were imputed using MICE, as described above.

The primary outcome was all-cause mortality within 30 days of RRT initiation.

### 3.2. Prediction Machine Learning Algorithms

Predicting mortality problem belongs to a classification topic in supervised machine learning. Four machine learning classification methods were applied in this study: logistic regression (LR), XGBoost, random forest (RF), and multilayer perceptron (MLP). We used grid search with tenfold cross-validation to find the best hyperparameters for all models. Our machine learning modeling strategy followed TRIPOD statement recommendations for the reporting of predictive models [23].

LR is the fundamental algorithm for machine learning development. In scikit-learn, the LR uses regularization by default. The advantage of regularization is to improve numerical stability.XGBoost [24] is an implementation of the gradient-boosted decision trees ensemble algorithm. The implementation of XGBoost is optimized for performance and provides the best available solutions in many fields. It reduces variance and bias by using multiple models and adjusting the subsequent trees by the errors the previous trees made.RF [25] is a bagging ensemble machine learning model that also includes several decision trees, but decisions made among trees are independent. It chooses the final model by voting for the most common class that reduces variance in decision trees. The advantages of RF are as follows: it is robust to overfitting and is more stable in high-dimensional data than other machine learning algorithms [26].MLP [27] is a well known supervised learning implementation in artificial neural networks. Typically, it consists of one input layer, one or more hidden layers, and one output layer. It solves high-dimensional classification problems by dealing with the interactions among variables.

## 4. Model Validation

Models were validated using two strategies (Figure 1): Using the first strategy to assess validation, we used the MIMIC dataset as a development model and assessed external validity using the eICU dataset, which was based on the higher severity of comorbidities and more complete records. Using the second strategy to build more robust models that could be applied across institutions, we pooled eICU and MIMIC datasets containing more diverse and heterogeneous data so that the trained models would generalize across different hospitals and then randomly split them into training and testing datasets at a ratio of 8:2. We performed grid searches with tenfold cross-validation to obtain the best parameters using the training dataset and then tested models on the independent testing dataset.

## 5. Statistical Analyses

We compared the baseline characteristics between the survival and death groups. Categorical variables were presented as proportions, and the mean with standard deviation or median with interquartile range was used to summarize the results for continuous variables. Numeric variables of clinical characteristics with normal distribution tested by the Kolmogorov–Smirnov test between the two groups were compared using the Student’s *t*-test. Non-normally distributed continuous variables were tested by a Mann–Whitney U test. A Chi-squared test was used to compare the differences in categorical variables.

The overall performance of the prediction models on validation was assessed by the calculation of the area under the receiver operating characteristic curve (AUC) and the associated 95% confidence interval (CI) using the roc_auc_score function of scikit-learn. Calibration was assessed using the Hosmer–Lemeshow test and by constructing calibration curves. The differences between model AUCs were pairwise-compared using the DeLong test (*p* < 0.05 was considered statistically significant). Sensitivity, specificity, positive (PPV) and negative (NPV) predicted values, and accuracy were calculated for evaluation of model performance. To evaluate the impact of features on our best model, we used the SHAP framework (available in the public domain at https://github.com/slundberg/shap (accessed on 5 February 2020) [28]. We used decision curve analysis to assess the net benefits of our best machine learning model, SOFA, nonrenal SOFA, and HELENICC scores. In the decision curve analysis, SOFA, nonrenal SOFA, and HELENICC scores were converted to a logistic regression using probability theory [12].

Machine learning algorithms and statistical analyses were performed using Python version 3.6, scikit-learn version 0.22.1, keras version 2.3.1, and R version 3.6.1.

## 6. Results

### 6.1. General Demographics

Of 3357 patients in the MIMIC and 8201 patients in the eICU databases who required dialysis therapy, 1129 and 2283, respectively, met the criteria for study inclusion (Figure 2).

Baseline demographics, comorbidities, vital signs, and laboratory values for patients in the two datasets are grouped by survival status (Appendix A). Overall, the cohorts from the group who died were older and had a lower percentage of black race, a longer ICU stay before dialysis therapy initiation, and a higher percentage of mechanical ventilation use. Appendix A reveals that the mortality rate and comorbidity are significantly different between the eICU and MIMIC datasets. The 30-day ICU mortality rate was 42.9% and 32.7% in the MIMIC and eICU datasets, respectively.

Table 1 shows the differences between the training and testing datasets of the pooled dataset. Regarding BUN, hemoglobin, and glucose, other variables were similar between the training and testing datasets.

### 6.2. SOFA, Nonrenal SOFA, and HELENICC Scores Performance in the MIMIC and eICU Datasets

The SOFA scores of the survival/death groups in the MIMIC and eICU cohorts were 8.7 ± 3.4/11.5 ± 3.6 (*p* < 0.001) and 10.6 ± 3.1/13.6 ± 3.2 (*p* < 0.001), respectively. The nonrenal SOFA scores of the survival/death groups in the MIMIC and eICU cohorts were 5.7 ± 3.3/8.8 ± 3.6 (*p* < 0.001) and 7.7 ± 3.2/11.1 ± 3.3 (*p* < 0.001), respectively. The performance of SOFA, nonrenal SOFA, and HELENICC scores in 30-day mortality prediction was modest in both datasets (Table 2). The nonrenal SOFA score performed better than the SOFA and HELENICC scores (0.728; 95% CI, 0.699–0.758 and 0.769; 95% CI, 0.749–0.789 in the MIMIC and eICU cohorts, respectively).

### 6.3. Machine Learning Algorithm Performance and Comparison with Other Predictive Models in the First Strategy

In the eICU (testing) dataset, the RF model achieved the highest AUC (0.816; 95% CI, 0.798–0.834) (Table 3), but there were no significant differences between those models (Appendix A). All four models performed significantly better than the SOFA, nonrenal SOFA, and HELENICC scores (*p* < 0.001). The Hosmer–Lemeshow test showed a poor model fit, except the MLP model. Figure 3 illustrates the ROC curves for our models, as well as for the SOFA, nonrenal SOFA, and HELENICC scores. Appendix A demonstrate the calibration curves of all models in the training and testing datasets.

### 6.4. Machine Learning Algorithm Performance and Comparison with Other Predictive Models in the Secondary Strategy

In the pooled dataset, there were 978 and 258 deaths in the training and testing datasets, respectively. The XGBoost model achieved the highest AUC value and accuracy (0.823; 95% CI, 0.791–0.854; 0.758) (Table 4). The MLP model performed worse than the other three models (Appendix A). The XGBoost, LR, and RF models showed no evidence of lack of fit with a Hosmer–Lemeshow test (*p* > 0.05) in the testing dataset. All four models performed significantly better than the SOFA, nonrenal SOFA, and HELENICC scores (*p* < 0.001). Figure 4 shows the ROC curves for all models. Appendix A demonstrate the calibration curves of all models in the training and testing datasets. The decision curve analysis showed that the net benefit of the XGBoost model was superior to the previous scoring systems (Figure 5).

### 6.5. Important Features of Machine Learning Algorithm and Results of Multivariable Logistic Regression Analysis

Figure 6a shows the top 10 important features of the XGBoost model that were calculated by SHAP value in the training datasets. In the 80% pooled dataset, older age, higher FiO_2_ and RR, lower creatinine and HCO_3_, increased anion gap, lower GCS, lower BP, vasopressor use, and decreased platelet count were associated with an increased mortality rate. The predictor ranks of all the models by the mean absolute value of SHAP are shown in Figure 6b. The results of multivariable logistic regression analysis using stepwise variable selection are shown in Appendix A and were similar to those of the XGBoost model.

## 7. Discussion

SOFA, nonrenal SOFA, and HELENICC scores have only modest predictive value of 30-day mortality for ICU patients with AKI requiring RRT. In the first strategy, we found that the machine learning models performed better than SOFA, nonrenal SOFA, and HELENICC scores when validating models using an external validation dataset (eICU dataset). In the second strategy, the XGBoost model showed reasonable performance and a sufficiently good fit (*p* = 0.22) in the heterogenous dataset. Decision curve analysis indicated that the XGBoost model improved the net benefit for predicting the 30-day mortality compared with SOFA, nonrenal SOFA, and HELENICC scores. The reasons for why the XGBoost model performed better may be related to the application of regularization and high flexibility to tune hyperparameters.

General severity of illness scores, which use clinical and laboratory variables at ICU admission or even sequential data, predict mortality well for all ICU patients, like APACHE, SAPS, and SOFA scores. However, they showed poor mortality prediction for AKI patients requiring RRT [10,11,12,13]. The models targeted specifically at this population including the HELENICC score, ATN study, and Cleveland Clinic score revealed good performance in predicting mortality (AUC = 0.82, 0.85, 0.81, respectively) [13,16,29]. Those models either lacked external validation, did not perform well during external validation [15], or focused on patients with specific conditions. Notably, some variables were not readily available in the clinical datasets we used, thus limiting our ability to make a direct comparison of ML models with these scores. Prior studies have predicted mortality using machine learning algorithms, although these did not center on RRT. Brajer et al. [30] revealed excellent performance using XGBoost to predict the in-hospital mortality of adults (AUC~0.85). Another study developed models for patients with influenza infection requiring ICU admission and found that XGBoost achieved the highest AUC (0.842) [31]. Kang et al. [12] applied machine learning algorithms to predict mortality in patients requiring CRRT and found that the RF model achieved the highest AUC (0.768). This was a retrospective study in one hospital (*n* = 1094) and lacked external validation. The performance of our models was reasonable high for all AKI patients requiring RRT, either using the eICU dataset as an external validation dataset or using the independent part of the pooled dataset as a testing dataset to provide results that are more generalizable.

In this study, we found that the performance of validating models was better when using the eICU dataset. This may be related to the patient characteristics of the two datasets. Patients in the MIMIC dataset had a higher mortality rate and more severe comorbidities than those in the eICU dataset (Appendix A). We speculate that this could be due to demographic differences, as the eICU dataset included data from multiple ICUs in rural areas while the MIMIC dataset contained data from a single ICU in an urban medical center. Another reason may be that the data distribution of MIMIC was more complicated to classify than that of eICU (Appendix A) using principal component analysis.

One challenge for medical researchers using machine learning algorithms is that it is difficult to assess or explain the individual contributing factors [32]. However, scientists are creating many advanced ways to make machine learning more transparent. We used the SHAP value to visualize the feature importance and determine the effect of different variables on the final output. SHAP offers not only the rank order of importance of variables but also how the variables impact the outcome, such as low creatinine associated with death risk. The results highly correlated with clinical outcome. Overall, the features generated by SHAP could be classified into hemodynamic status, central nerve system, coagulation, respiratory systems, kidney-related features, and age. The SHAP results were like the SOFA score parameters, but the performance of our models outperformed the SOFA score. That may be related to the loss of information when categorizing data, inaccurately allocating creatinine score for those AKI patients using the SOFA score, and capturing nonlinear interactions from high complexity data using machine learning algorithms. In this study, the XGBoost model used more and different variables, such as age and anion gap, which may lead to better performance than the SOFA and HELENICC scores. Besides, we retrained a new XGBoost model only using the top 10 features generated by SHAP and still achieved a good AUC (0.818; 95% CI: 0.786–0.849) in the testing dataset, which allows clinical physicians to use the model by inputting only 10 data points, can minimize the burden on them, and limit non-use in the case of missing data on a larger number of variables.

Given its improved performance over traditional severity-of-illness scoring measures, such a model or tool could potentially be used and further refined for several potential applications. For example, given its relatively high negative predictive value, it might help to enrich clinical trials for a targeted risk profile of patients. Another potential advantage of our model is that utilizes data that is easily available and routinely collected in clinical practice. This is a distinct advantage over some other prior risk scores, such as ATN score, that have been used in this population, as those included data collected as part of a research study and may not be available clinically. Having a model that can be calculated in real-time will allow clinicians to have prognostic data and help them have informed shared decision-making with the patient and their caregivers to decide whether to initiate RRT. This will also allow physicians to discuss the overall aggressiveness of care and help with medical decision-making

The strengths of the study include large sample size, external validation, curated datasets representing heterogeneous ICU populations, and the use of routinely collected clinical data. This study has several limitations. The MIMIC III dataset is old, and practice patterns may have changed. This dataset is only limited to labs when the patient is in the ICU, so it is possible that we may have missed AKI by creatinine values if they were admitted to the hospital in a non-ICU setting prior to their ICU admission. Although our datasets included a large amount of collected clinical data, some data had to be excluded due to poor-quality data recording or missing data. Using MICE to impute data may reduce predictive power. Due to many missing values, we were unable to calculate SAPS and APACHE at RTT start time. Thus, we could not make a head-to-head comparison of the performance of SAPS or APACHE with our models. In addition, the datasets did not have variables to allow for comparison with other scores that have looked specifically at this population, such as the ATN study score, which included research data. Moreover, the variables only capture data collected in the hospitals, and we are unable to capture patient mortality out of the hospitals in the eICU dataset. Our sample size may not be large enough for machine learning to have better performance. The interpretation of results should only focus on specific patients, since we excluded all transplant and advanced CKD patients in the United States. Finally, the causal relationship between the top 10 features of the XGBoost model and 30-day mortality was not clearly explored.

## 8. Conclusions

All machine learning models had a reasonable performance and were superior to the SOFA, nonrenal SOFA, and HELENICC scores in predicting 30-day mortality for AKI patients requiring RRT. XGBoost provided the highest performance in this study. Further prospective research is needed to validate these results prospectively and explore how they can be integrated into clinical decision-making.

## Figures and Tables

**Figure 1 jcm-11-05289-f001:**
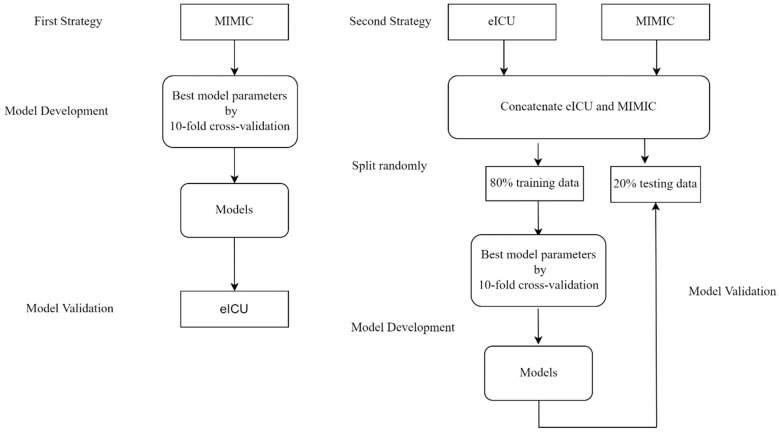
Model development and validation flowchart.

**Figure 2 jcm-11-05289-f002:**
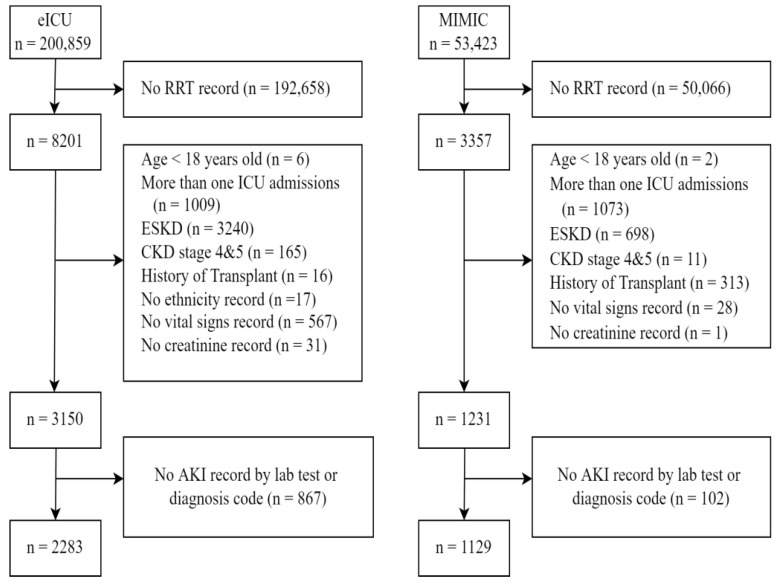
Participant flow diagram. *n* is patient unit encounter.

**Figure 3 jcm-11-05289-f003:**
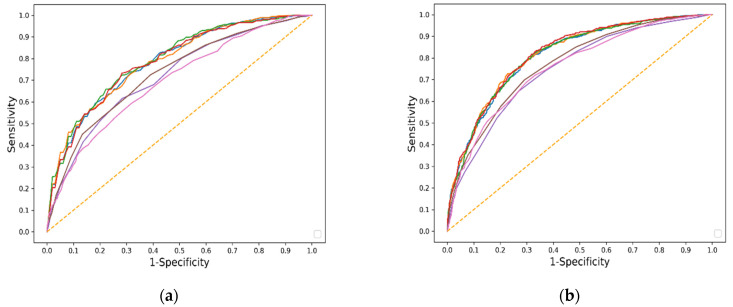
Area under operating characteristic (ROC) curves for prediction models trained in MIMIC (**a**) and tested in eICU (**b**). Red, RF (0.783, 0.816); Green, XGBoost (0.793, 0.812); Blue, MLP (0.785, 0.810); Orange, LR (0.786, 0.815); Brown, nonrenal SOFA score (0.728, 0.769); Purple, SOFA score (0.717, 0.749); Pink, HELENICC score (0.694, 0.756).

**Figure 4 jcm-11-05289-f004:**
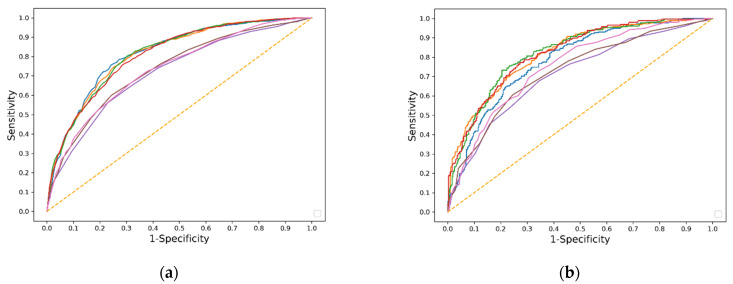
Area under operating characteristic (ROC) curves for prediction models trained in 80% pooled data (**a**) and tested in 20% pooled data (**b**). Red, RF (0.809, 0.821); Green, XGBoost (0.814, 0.823); Blue, MLP (0.818, 0.784); Orange, LR (0.814, 0.819); Brown, nonrenal SOFA score (0.735, 0.726); Purple, SOFA score (0.718, 0.710); Pink, HELENICC score (0.735, 0.752).

**Figure 5 jcm-11-05289-f005:**
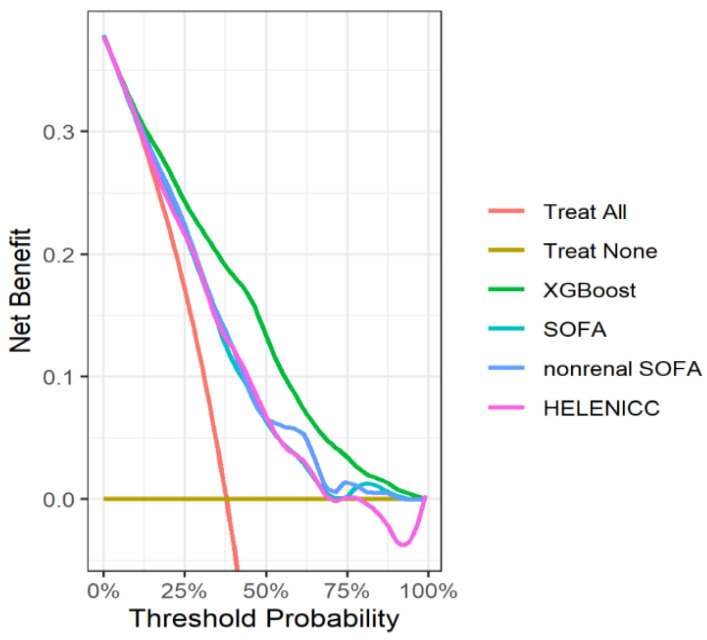
Decision curve analysis of random forest, SOFA, nonrenal SOFA, and HELENICC for 30-day mortality prediction.

**Figure 6 jcm-11-05289-f006:**
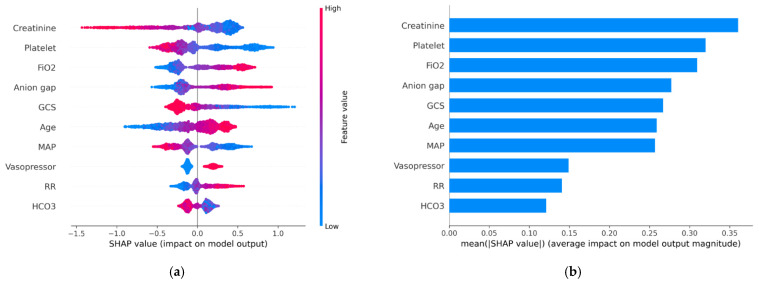
(**a**) Top 10 important features of the XGBoost model by SHAP value. The X-axis shows the distribution of the SHAP value, which reveals how much it impacts the outcome. The Y-axis is the feature value. The color represents the positive or negative contribution to outcome (red prone to death, blue prone to survival). (**b**) Ranks of the predictors by the mean absolute value of the SHAP values.

**Table 1 jcm-11-05289-t001:** Baseline characteristics of the patients requiring renal replacement therapy between the training and testing datasets of pooled data.

Variables	Training Dataset	Testing Dataset	*p* Value
Number of patients	2729	683	
Death %	35.8%	37.8%	0.369
**Demographics**			
Age, years	62.9 ± 15.0	62.7 ± 14.9	0.793
Male sex, %	59.5%	59.7%	0.962
Black race, %	16.8%	18%	0.149
**Comorbidities (%)**			
Diabetes mellitus	21.6%	21.1%	0.800
Hypertension	25.9%	25.0%	0.677
CHF	24.8%	26.4%	0.445
CKD	17.5%	16.7%	0.667
Malignancy	6.7%	6.9%	0.910
Liver cirrhosis	10.6%	9.7%	0.523
Days of ICU stay before RRT initiation	2.9 ± 4.8	3.1 ± 5.1	0.223
Diuretics, %	14.4%	12.0%	0.126
Vasopressor, %	36.5%	38.4%	0.381
Mechanical ventilation, %	72.4%	74.7%	<0.254
**Laboratory variables**			
BUN (mg/dL)	56.0 (36.0–84.0)	61.0 (38.0–89.0)	0.01 *
FiO_2_ (%)	49.5 ± 26.8	48.7 ± 26.3	0.501
HCO_3_ (mmol/L)	20.5 ± 5.7	20.3 ± 5.8	0.464
Hgb (mg/dL)	9.6 (8.5–10.8)	9.8 (8.6–11.1)	0.02 *
O_2_ Sat (%)	93.7 ± 8.0	93.6 ± 8.0	0.767
WBC count (×1000/μL)	15.8 ± 26.2	15.8 ± 14.6	0.976
Anion gap (mmol/L)	16.5 ± 6.7	16.9 ± 6.4	0.155
Calcium (mg/dL)	8.2 ± 1.1	8.2 ± 1.1	0.959
Creatinine (mg/dL)	4.6 ± 3.2	4.6 ± 2.9	0.713
Glucose (mg/dL)	146.0 ± 65.1	152.8 ± 87.7	0.024
Platelet count (×1000/μL)	182.1 ± 115.1	175.7 ± 106	0.193
Potassium (mmol/L)	4.7 ± 1.0	4.7 ± 1.0	0.349
Sodium (mmol/L)	137.4 ± 6.1	137.5 ± 6.1	0.537
GCS score	11.1 ± 4.1	11.0 ± 4.0	0.381
MAP (mmHg)	76.2 ± 15.0	75.3 ± 14.6	0.170
HR (beats per minute)	89.4 ± 18.4	90.0 ± 18.7	0.444
RR (breaths per minute)	20.9 ± 5.6	21.1 ± 5.4	0.374

Abbreviations: CHF, congestive heart failure; CKD, chronic kidney disease; BUN, blood urea nitrogen; FiO_2_, fraction of inspired oxygen; Hgb, hemoglobin; WBC, white blood cell; GCS, Glasgow Coma Scale; HR, heart rate; MAP, mean arterial pressure; RR, respiratory rate; SI, shock index; ICU, intensive care unit; RRT, renal replacement therapy; CRRT, continuous renal replacement therapy; IHD, intermittent hemodialysis; MV, mechanical ventilation. Data are expressed as *n* (%) for categorical data and as mean ± standard deviation or median (interquartile range) for continuous data. * Mann–Whitney U test.

**Table 2 jcm-11-05289-t002:** Performance of the SOFA, nonrenal SOFA, and HELENICC scores 1 day before the beginning of dialysis in the MIMIC and eICU datasets.

Dataset	MIMIC			eICU		
Model	SOFA	Nonrenal SOFA	HELENICC	SOFA	Nonrenal SOFA	HELENICC
AUC	0.717	0.728	0.694	0.749	0.769	0.756
95% CI	0.687–0.747	0.699–0.758	0.664–0.752	0.728–0.770	0.749–0.789	0.735–0.776
Sensitivity	0.514	0.528	0.401	0.372	0.446	0.341
Specificity	0.798	0.792	0.845	0.884	0.868	0.914
PPV	0.656	0.656	0.659	0.612	0.625	0.662
NPV	0.687	0.691	0.653	0.741	0.761	0.738
Accuracy	0.676	0.679	0.654	0.715	0.729	0.725

Abbreviations: PPV, positive predictive value; NPV, negative predictive value.

**Table 3 jcm-11-05289-t003:** Model performance measures in the MIMIC and eICU datasets.

Training Dataset	MIMIC			
Model	LR	XGBoost	RF	MLP
AUC	0.786	0.793	0.783	0.785
95% CI	0.752–0.820	0.760–0.826	0.743–0.822	0.752–0.819
Sensitivity	0.578	0.619	0.621	0.617
Specificity	0.809	0.803	0.800	0.779
PPV	0.694	0.702	0.700	0.678
NPV	0.719	0.737	0.738	0.731
Accuracy	0.710	0.724	0.723	0.710
Hosmer-Lemeshow test	<0.05	0.02	<0.05	0.44
Testing Dataset	eICU			
Model	LR	XGBoost	RF	MLP
AUC	0.815	0.812	0.816	0.810
95% CI	0.797–0.833	0.794–0.830	0.798–0.834	0.792–0.828
Sensitivity	0.440	0.488	0.595	0.489
Specificity	0.905	0.892	0.837	0.885
PPV	0.695	0.691	0.642	0.677
NPV	0.767	0.780	0.808	0.779
Accuracy	0.752	0.759	0.757	0.755
Hosmer-Lemeshow test	<0.05	<0.05	<0.05	0.29

**Table 4 jcm-11-05289-t004:** Model performance measures in the pooled dataset.

Training Dataset	80% Pooled Data			
Model	LR	XGBoost	RF	MLP
AUC	0.814	0.814	0.809	0.818
95% CI	0.797–0.831	0.800–0.828	0.796–0.822	0.802–0.833
Sensitivity	0.574	0.584	0.556	0.644
Specificity	0.845	0.833	0.853	0.825
PPV	0.674	0.662	0.679	0.672
NPV	0.780	0.782	0.774	0.805
Accuracy	0.748	0.744	0.746	0.760
Hosmer-Lemeshow test	<0.05	0.02	0.08	<0.05
Testing Dataset	20% Pooled Data			
Model	LR	XGBoost	RF	MLP
AUC	0.819	0.823	0.821	0.784
95% CI	0.787–0.851	0.791–0.854	0.790–0.852	0.750–0.817
Sensitivity	0.620	0.635	0.562	0.662
Specificity	0.804	0.832	0.863	0.785
PPV	0.658	0.697	0.714	0.652
NPV	0.777	0.790	0.764	0.793
Accuracy	0.734	0.758	0.749	0.739
Hosmer-Lemeshow test	0.11	0.22	0.17	<0.05

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
