# Peer review of "Predicting Mortality Using Machine Learning Algorithms in Patients Who Require Renal Replacement Therapy in the Critical Care Unit"

_jcm, 2022, doi:10.3390/jcm11185289_

Round 1
Reviewer 1 Report
To the authors
Thank you for this well written and presented paper regarding the comparison of different machine learning algorithms with “classical scores” for predicting mortality in ICU-RRT patients, structured according to TRIPOD Guidelines. Supporting clinical decisions with computerized risk prediction is definitely an interesting, actual and relevant topic in medicine and especially in those critically ill patients requiring RRT during ICU treatment.
Nevertheless, there are a few concerns, which I like to address. The main point is that it is not clear to me whether you are presenting a proof of concept or an approach to support a particular clinical decision. If it is a proof of (machine learning) concept, I have some reservations about your work. There is already evidence in the literature that machine learning algorithms (XGBoost in particular) can be superior to classical scores, which, in part, you also refer to. Therefore, although your data are well prepared and presented, in this sense they bring little new insights.
Otherwise, if your goal is to present a new score to support physicians in clinical decisions regarding ICU patients with RRT, I recommend to highlight this more clearly to the reader. E.g. what is the advantage of using data one day prior to RRT initialization? I understand that some systemic problems exist regarding the data quality at the day of RRT initialization, but may there some potential clinical benefits regarding your chosen time point? What is the new clinical impact here compared to presented machine learning algorithm, which also revealed XGBoost as superior? What about general mortality prediction in ICU? What is the beneficial clinical implication compared to the mentioned existing predictive models and how would you implement your algorithm in clinical praxis?
Also a further aspect to mention, in ICU therapy the question arises as to the best possible time to initiate RRT or in which cases RRT should be initiated at all, which is also emphasized by some of your mentioned references (e.g. Zarbock et al.). What may be the advantage of mortality prediction in already initiated RRT compared to those open questions and may by your approach also helpful in solving these?
Again, I like your approach and machine learning is pretty much an important future pathway, but you should convince the reader of the potential positive clinical benefits and thus also what is new about this approach.
Beside this, some minor points are:
- pooled results using MICE contains limitations to discuss… validation of the whole dataset and impact on predictive power e.g.
- What about renal failure without RRT ? Which day/time point of ICU therapy was the “day before RRT”. What about the mortality prediction in AKI patients at this time point without following RRT?
- In my opinion it is difficult to count eICU data set as an external validation. Where the algorithm learning solely due to MIMIC III data? E.g. what about cross check within a complete different health care system in another country? Please comment/discuss this.
- I would recommend to report a p>0.05 in Hosmer-Lemeshow test as a hind for absence of poor calibration. This is not equal to a proof for a well calibrated model.
- When reporting T-test, prior analyzation of normal distribution should be performed. Please state, if this was done for your data.
- Please introduce all used abbreviations within the manuscript e.g. SHAP and also within the tables (e.g laboratory findings)
- Why using FiO2 reported from ABG and not from the Respirator? ABG FiO2 is often manually entered and therefore subject to errors. Please comment or mention in limitations.
- What about implementing the main diagnosis within the risk prediction? It is conceivable that this would have a further impact on accuracy. Can you comment on this?
- were missing values also handled by XGBoost or solely by MICE, so that XBoost only handled complete data sets?
- please revise carefully for spelling and punctuation (E.g. ll. 297 “bedsides”)
- Please add Source: Johnson, A., Pollard, T., & Mark, R. (2016). MIMIC-III Clinical Database (version 1.4). PhysioNet. https://doi.org/10.13026/C2XW26. Page 5, Row 94
- Tables and figures in the supplement have other labeling then at the end of the manuscript, please check.
Author Response
Response to Referee: 1
Comments to the Author
Thank you for this well written and presented paper regarding the comparison of different machine learning algorithms with “classical scores” for predicting mortality in ICU-RRT patients, structured according to TRIPOD Guidelines. Supporting clinical decisions with computerized risk prediction is definitely an interesting, actual and relevant topic in medicine and especially in those critically ill patients requiring RRT during ICU treatment.
- The main point is that it is not clear to me whether you are presenting a proof of concept or an approach to support a particular clinical decision. If it is a proof of (machine learning) concept, I have some reservations about your work. There is already evidence in the literature that machine learning algorithms (XGBoost in particular) can be superior to classical scores, which, in part, you also refer to. Therefore, although your data are well prepared and presented, in this sense they bring little new insights. Otherwise, if your goal is to present a new score to support physicians in clinical decisions regarding ICU patients with RRT, I recommend to highlight this more clearly to the reader. E.g. what is the advantage of using data one day prior to RRT initialization? I understand that some systemic problems exist regarding the data quality at the day of RRT initialization, but may there some potential clinical benefits regarding your chosen time point? What is the new clinical impact here compared to presented machine learning algorithm, which also revealed XGBoost as superior? What about general mortality prediction in ICU? What is the beneficial clinical implication compared to the mentioned existing predictive models and how would you implement your algorithm in clinical praxis? Also a further aspect to mention, in ICU therapy the question arises as to the best possible time to initiate RRT or in which cases RRT should be initiated at all, which is also emphasized by some of your mentioned references (e.g. Zarbock et al.). What may be the advantage of mortality prediction in already initiated RRT compared to those open questions and may by your approach also helpful in solving these?
Response:
We would like to thank the reviewer for the opportunity to clarify this. Our goal is the latter – to present a new score to help with clinical decision making in the ICU before RRT is initiated. Often times clinicians lack the objective data to facilitate prognostic discussions with family. Using data 1 day prior to RRT, we show high performance on predictive accuracy of this score. The advantage of our score is that utilizes data that is easily available and routinely collected in clinical practice. This is a distinct advantage over some other prior risk scores such as ATN score that have been used in this population, as those included data collected as part of research study and may not be available clinically. Having a score that can be calculated in real-time will allow clinicians to have prognostic data and help them have informed shared decision-making with the patient and their caregivers to decide whether to initiate RRT. This will also allow physicians to discuss overall aggressiveness of care and help with medical decision making. We have expanded this in the discussion section. (line 307 page 14)
- Pooled results using MICE contains limitations to discuss
Response:
Response:
We fully agree with the reviewer that using MICE to impute data is a limitation since we used a medical database which usually has missing data. We revised this in manuscript (line 321, page 15).
- What about renal failure without RRT? Which day/time point of ICU therapy was the “day before RRT”. What about the mortality prediction in AKI patients at this time point without following RRT?
Response:
We thank the reviewer for allowing us to explain more. Since we excluded patients who didn’t undergo RRT, we don’t predict mortality in this patient group. The definition of “day before RRT” in this study was 24 hours before RRT was initiated. We have clarified this in the manuscript. (line 118 page 6).
- In my opinion it is difficult to count eICU data set as an external validation. Where the algorithm learning solely due to MIMIC III data? E.g. what about cross check within a complete different health care system in another country? Please comment/discuss this.
Response:
We thank the reviewer for allowing us to explain more. Although these two databases are from U.S., there is no overlap of patients. (line 98, page 5) That would be the next step for our team – to validate in a heterogenous multi-national cohort. However, before doing that we wanted to validate our model in an external dataset which provided heterogenous patient population from multiple hospitals in the US. (line 171 page 8)
- I would recommend to report a p>0.05 in Hosmer-Lemeshow test as a hind for absence of poor calibration. This is not equal to a proof for a well calibrated model.
Response:
We agree with the reviewer that Homer-Lemeshow test can indicate absence of poor calibration, which is not equivalent to proof of well calibrated model. We have changed the wording in our manuscript to reflect this. (line 55 page 3, line 235 page11)
- Please introduce all used abbreviations within the manuscript e.g. SHAP and also within the tables (e.g laboratory findings)
Response:
We thank the reviewer for allowing us to explain more. We have presented the full form of all abbreviations in manuscript and tables as applicable.
- When reporting T-test, prior analyzation of normal distribution should be performed. Please state, if this was done for your data.
Response: We appreciate the reviewer’s valuable comment. We revised it in the Method section and all tables. (line 177 page 9)
- Why using FiO2 reported from ABG and not from the Respirator? ABG FiO2 is often manually entered and therefore subject to errors. Please comment or mention in limitations.
Response: We appreciate the reviewer’s valuable suggestions. FiO2 was got from respiratory Charting in eICU and ABG in MIMIC. We revised it. (line 124 page 6)
- What about implementing the main diagnosis within the risk prediction? It is conceivable that this would have a further impact on accuracy. Can you comment on this?
Response: We appreciate the reviewer’s valuable comments. Often the main diagnosis is not captured well in structured data in EHR. It needs natural language processing to parse free text and that would be our next step.
- Were missing values also handled by XGBoost or solely by MICE, so that XGBoost only handled complete data sets?
Response: We thank the reviewer for allowing us to explain more. The missing values were completed handled by MICE and then the imputed data were used to build models. We have added this to the methods section. (line 132 page7)
- Please revise carefully for spelling and punctuation (E.g. ll. 297 “bedsides”)
Response: We revised it (line 299, page 14)
- Please add Source: Johnson, A., Pollard, T., & Mark, R. (2016). MIMIC-III Clinical Database (version 1.4). PhysioNet. https://doi.org/10.13026/C2XW26. Page 5, Row 94
Response:. We revised it and added eICU reference. (line 95 and 98 page 5)
- Tables and figures in the supplement have other labeling then at the end of the manuscript, please check.
Response: We thank the reviewer’s valuable comments. We revised all the labeling.

Reviewer 2 Report
Introduction
- Why is mortality high in patients with AKI receiving RRT
- Why do the current scoring mechanisms fail to provide a reliable prediction accuracy and why is it not reliable for the AKI population?
- In what time window do you define RRT initiation? As AKI is rapidly progressing and RRT is set up, how do novel models score better upon RRT initiation compared to other models? Elaborate as well on which moments in time the existing scores are defined and if they are repeated during the ICU stay
Methods
- Elaborate on why mortality is assessed at 30 days? Why is this time line clinically relevant?
- What about the time patients were on RRT and characteristics of RRT itself?
- Why was eGFR not taken into account?
- It is unclear at which time points the comparison scores were taken
- Why was the reason of hospitalisation not included?
- Why was the reason of mortality not included?
- Elaborate on the decision to use the two presented databases and why they were used as separate databases; moreover, elaborate in the methods section you will be mostly comparing deceased vs survival patients
- What about the patients that deceased prior to 30-days? Were they included?
- Were patients with only 1 kidney included?
- How was hypertension defined (as well as the other comorbidities)? In other words, was hypertension (and the other comorbidities) included as relevant medical history or an active/acquired diagnosis for which the patient needed treatment?
- Why was liver cirrhosis chosen as a comorbidity?
- At what frequency were MAP, GCS and the vital signs measured and how were they incorporated as features into the models?
- Clarify that the models you are using are ad hoc models and not real-time models
- Elaborate more in detail which parameters were included in the ML models
- Elaborate on the choice why the MIMIC dataset was used for training and how bias was avoided
- Why did you only use your top 10 of features to train the models?
Results
- How was the proportion of men and women?
- What about the overall LOS?
- Can you report the non-renal SOFA scores for both databases (survival/deceased)
- line 241: were these features dependent or independent of each other?
Discussion
- elaborate in what the feature selection differs in the XGBoost model vs the standard of care models (SOFA, non-renal, HELENICC)
- line 263: can you provide examples on the specific conditions and why these models were not sufficient?
- Discussion should be enriched on the clinical value of your research
- 312: you state that you could not calculate the SAPS and APACHE score for every patient, indicating that the non-renal SAPS score also not can be calculated ïƒ yet you state in your results that your models work better compared to SAPS, etc. ïƒ it seems that this is an incorrect statement to make. In addition, I did not see any results in comparison to the APACHE score (if no results are linked to this, don’t mention this score as it has no added value and creates confusion)
Author Response
Response to Referee: 2
- Why is mortality high in patients with AKI receiving RRT ?
Response:
We thank the reviewer for allowing us to explain more. The main risks for patient with AKI receiving RRT are older age, sepsis, decompensated heart failure, liver failure, and higher severity of illness, which are associated with high mortality. (Tandukar, Srijan, and Paul M. Palevsky. "Continuous renal replacement therapy: who, when, why, and how." Chest 155.3 (2019): 626-638.)
- Why do the current scoring mechanisms fail to provide a reliable prediction accuracy and why is it not reliable for the AKI population?
Response:
We appreciate the reviewer’s critical comments. Those scoring systems are designed for general hospitalization patients and don’t focus on AKI population. For example, low creatinine may be associated with high mortality in AKI (Clinical Journal of the American Society of Nephrology 2011, 6, 2114-2120), but higher mortality risk was defined as higher creatinine level in general scoring systems. In other words, we need specific mortality model for AKI population to make better prediction.
- In what time window do you define RRT initiation? As AKI is rapidly progressing and RRT is set up, how do novel models score better upon RRT initiation compared to other models? Elaborate as well on which moments in time the existing scores are defined and if they are repeated during the ICU stay
Response:
We thank the reviewer for allowing us to explain more. For RRT initiation, we tried to search any key words related with dialysis, like CVVH, dialysis, hemodialysis, etc., by the program released by official site of MIMIC. (https://github.com/MIT-LCP/mimic-code/blob/main/mimic-iii/concepts/pivot/pivoted_rrt.sql) In eICU, we searched “dialysis” in treatment table by official suggestion in discussion forum. (https://github.com/MIT-LCP/eicu-code/issues/83). Our models used routinely collected clinical data, focused on patients AKI requiring dialysis, and had abilities to handle complex data automatically. To our knowledge, those existing scores for AKI, like HELENICC or ATN score, used the variables at the time of starting dialysis because they were prospective research studies and they were not repeated during the ICU stay. We used data within 24 hours before RRT because that provides a clinical window of opportunity to have prognostic discussions and shared decision making with the family regarding RRT
- Elaborate on why mortality is assessed at 30 days? Why is this time line clinically relevant?
Response:
We thank the reviewer for allowing us to explain more. 30-days is standardized period which was chosen to ensure a fair assessment of all hospitals and to prevent differences in transfer rates or variations in length of stay from affecting the measurement. This approach is often used and reported in evaluation of programs and procedures and is part of clinical quality metrics (Krumholz, Harlan M., and Sharon-Lise T. Normand. "Public reporting of 30-day mortality for patients hospitalized with acute myocardial infarction and heart failure." Circulation 118.13 (2008): 1394-1397.) .
- What about the time patients were on RRT and characteristics of RRT itself?
Response:
We thank the reviewer for allowing us to explain more. We didn’t know the actual indications for patient with AKI requiring RRT because we used medical database retrospectively. The conventional indications for RRT are refractory acidosis, hyperkalemia, uremia, oliguria/anuria, and volume overload. (Vaara, Suvi T., et al. "Timing of RRT based on the presence of conventional indications." Clinical Journal of the American Society of Nephrology 9.9 (2014): 1577-1585) There are two popular RRT modalities in ICU, intermittent hemodialysis (HD) and continuous renal replacement therapy (CRRT). The choice of which modality to use is based on patient’s clinical condition but often is also dictated by other factors such as resource availability, expertise and comfort level of the ICU and nephrology teams. Given the variation in these external factors across different hospital systems, and lack of their availability in database, we did not include these in our models. We do not believe that these variables would have changed our results. In this study, the proportion of HD/CRRT are: 63.3%/36.7% in survival group, 40.5%/59.5% in death group in MIMIC; 80.7%/19.3% in survival group, 55.8%/44.2% in death group in eICU.
- Why was eGFR not taken into account?
Response:
We thank the reviewer for allowing us to explain more. Serum creatinine and eGFR are delayed and unreliable indicators of AKI, thus we did not include them. (Curr Opin Pediatr. 2011 Apr; 23(2): 194–200.)
- It is unclear at which time points the comparison scores were taken.
Response:
We thank the reviewer for allowing us to explain more. All the data used in our study was within 24 hours before RRT
- Why was the reason of hospitalization not included? Why was the reason of mortality not included?
Response:
We thank the reviewer for allowing us to explain more. The reason of mortality was not clearly mentioned in database. Besides, according to MIMIC III document, the reason of hospitalization can be very clear or quite vague (https://mimic.mit.edu/docs/iii/tables/admissions/). To ascertain the correct reason, we would need to do free text analysis which was beyond the scope of the current project. Additionally, we do not believe that including the reason would have changed our results as most patients in ICU with severe AKI requiring RRT will have multi-organ failure and similar physiological parameters (which were included in our models).
- Elaborate on the decision to use the two presented databases and why they were used as separate databases; moreover, elaborate in the methods section you will be mostly comparing deceased vs survival patients
Response:
We thank the reviewer for allowing us to explain more. The reasons why we chose those two databases are that they are two large, public ICU database in the U.S. And we used them separately because we wanted to prove if our models could work using an external validation dataset. The outcome of the models was all-cause mortality within 30 days of RRT initiation (line 141 page 7), thus characteristics of those surviving vs dead at 30-days are shown in Supplement Table S3 and S4.
- What about the patients that deceased prior to 30-days? Were they included?
Response:
As we have mentioned in the primary outcome all patients who deceased within 30-days of RRT initiation were included.
- Were patients with only 1 kidney included?
Response:
No matter how many kidneys patients have, we included patients who met inclusion criteria.
- How was hypertension defined (as well as the other comorbidities)? In other words, was hypertension (and the other comorbidities) included as relevant medical history or an active/acquired diagnosis for which the patient needed treatment?
Response:
As we have described in our methods section and Supplementary Table S1, all comorbidities were defined by ICD-9 (line 119 page 6).
- Why was liver cirrhosis chosen as a comorbidity?
Response:
We thank the reviewer for allowing us to explain more. AKI is one of the most severe complications of liver cirrhosis. The overall incidence of AKI in cirrhotic patients was 53.9%, and the overall hospital mortality was 28.4%. (Hepatology 57.2 (2013): 753-762., BMC nephrology 19.1 (2018): 1-8) So we included liver cirrhosis as an important variable for mortality prediction.
- At what frequency were MAP, GCS and the vital signs measured and how were they incorporated as features into the models?
Response:
We thank the reviewer for allowing us to explain more. The frequency of checking vital sings, GCS, and MAP is usually about 1 hour or 2 hours in ICU. But it depends on patient condition, it may be measured more frequent if patient is unstable, and other external factors such as staffing. In our study, we used mean values of all variables recorded within 24 hours before RRT initiation. (line 121 page 6)
- Clarify that the models you are using are ad hoc models and not real-time models
Response:
We thank the reviewer for allowing us to explain more. The goal of these models is to be available to guide clinical practice and decision making in the ICU and can be run ad-hoc using real time data. We have added this to the intro section. (line 87 page 5)
- Elaborate more in detail which parameters were included in the ML models
Response:
We thank the reviewer for allowing us to explain more. The variables used in this study are already described in Table1.
- Elaborate on the choice why the MIMIC dataset was used for training and how bias was avoided
Response:
We thank the reviewer for allowing us to explain more. Why we chose the MIMIC dataset as training dataset was MIMIC dataset is a large, open database that has been widely used. The advantage is that it includes data from a large tertiary medical center, and reflects higher acuity and quality of care. To minimize bias, we excluded patient with no AKI records, advanced CKD history, and no vital signs data. And variables with >30% missing values were also excluded.
- Why did you only use your top 10 of features to train the models?
Response:
We thank the reviewer for allowing us to explain more. Since the goal of our model is to be available for clinical decision making, we balanced the need for a well performing model with ease of use keeping in mind that busy physicians in the ICU are not likely going to use a score that is too cumbersome to calculate. Our model retained good performance with only using the top 10 features (line 299, page14). This will allow the clinical physicians to use the score by inputting only 10 data points and can minimize burden on them, and limit non-use in case of missing data on a larger number of variables.
- How was the proportion of men and women?
Response:
Please see patient characteristics detailed in Supplement Table 3 and 4 which show male/female in survival/death group was 59.7%/40.3%, 65.1%/34.9% in MIMIC, and 59%/41%, 57.2%/42.8% in eICU.
- What about the overall LOS?
Response:
We thank the reviewer for allowing us to explain more. The LOS (length of stay at hospital, days) of MIMIC was 15.83±11.97 and that of eICU was 12.06±9.91
- Can you report the non-renal SOFA scores for both databases (survival/deceased)
Response:
We thank the reviewer for allowing us to explain more. We revised in our manuscript. The SOFA scores of the survival/death groups in the MIMIC and eICU cohorts were 8.7 ± 3.4/11.5 ± 3.6 (P < 0.001) and 10.6 ± 3.1/13.6 ± 3.2 (P < 0.001), respectively. The non-renal SOFA score in survival/death group were 5.7±3.3/8.8±3.6 in MIMIC and 7.7±3.2/11.1±3.3 in eICU. (line 213, page10)
- line 241: were these features dependent or independent of each other?
Response:
We thank the reviewer for allowing us to explain more. SHAP values share credit for the interactions among each of the features. They should be dependent of each other.
- Elaborate in what the feature selection differs in the XGBoost model vs the standard of care models (SOFA, non-renal, HELENICC)
Response:
We thank the reviewer for allowing us to explain more. SOFA score was created by ESICM organized a consensus meeting in Paris in October 1994. ( Vincent, J-L., et al. "The SOFA (Sepsis-related Organ Failure Assessment) score to describe organ dysfunction/failure." (1996): 707-710.) HELENICC was found by logistic regression. (BMC Anesthesiol. 2017 Feb 7;17(1):21) XGBoost model is a gradient tree boosting algorithm based on decision tree. It tries to fit the new predictor to the residual errors made by the previous predictor under the gradient boosting framework. (Materials 2021, 14(4), 713)
- line 263: can you provide examples on the specific conditions and why these models were not sufficient?
Response:
We thank the reviewer for allowing us to explain more. HELENICC score was designed for septic AKI patients and ATN score was designed for patients had AKI attributable to acute tubular necrosis. Those models either lacked external validation, did not perform well during external validation, or were developed using research data that is not collected in routine clinical practice. (line 263 page 13) Our study was more general for all patient with AKI requiring RRT, using routine clinical data and also had an external validation.
- Discussion should be enriched on the clinical value of your research
Response:
We thank the reviewer for allowing us to explain more. The advantage of our score is that utilizes data that is easily available and routinely collected in clinical practice. This is a distinct advantage over some other prior risk scores such as ATN score that have been used in this population, as those included data collected as part of research study and may not be available clinically. Having a score that can be calculated in real-time will allow clinicians to have prognostic data and help them have informed shared decision-making with the patient and their caregivers to decide whether to initiate RRT. This will also allow physicians to discuss overall aggressiveness of care and help with medical decision making. We have expanded this in the discussion section. (line 307 page 14)
- 312: You state that you could not calculate the SAPS and APACHE score for every patient, indicating that the non-renal SAPS score also not can be calculated yet you state in your results that your models work better compared to SAPS, etc. ïƒ it seems that this is an incorrect statement to make. In addition, I did not see any results in comparison to the APACHE score (if no results are linked to this, don’t mention this score as it has no added value and creates confusion)
Response:
We thank the reviewer for allowing us to explain more. In this study, we compared our models with SOFA, nonrenal SOFA, and HELENICC scores. SAPS and APACHE are general severity of illness scores have been used to predict ICU mortality. We mentioned that we could not make a head-to-head comparison of the performance of SAPS or APACHE with our models because variables are unavailable (line 323, page 15).

Reviewer 3 Report
Thank you for the opportunity to review the manuscript titles “Predicting Mortality Using Machine Learning Algorithms in Patients Who Require Renal Replacement Therapy in the Critical Care Unit”. The authors attempted to study the value of machine learning models in predicting mortality in patients receiving renal replacement therapy in the Critical care unit for acute renal injury.
The commonly known scores have been shown to be less accurate in patients requiring RRT in the ICU, therefore the authors have elected to pursue the study of ML utilization for prognostic model development for this purpose. However, we find the methodology was not optimal enough to answer the question. The comparator scores were known to have fair performance, missing data management and methodology of model variables’ selection along with development and testing strategies are some of many to be commented on, we will elaborate further below.
The use of two strategies for models’ development using two highly variable dataset did not add value to the methodology. In the first strategy, it is not clear why the model development used the older, single-center, and smaller dataset while testing was on the larger multi-center eICU dataset. In addition, we believe reporting the results of both strategies did not contribute to the aim on the study rather added confusion on my part as a reader. Therefore, despite utilizing the same variables, the two versions of every model of the 4 ML developed by the two strategies were different regarding the ranking of the top 10 predictors along with their effect on outcome (Figure S6). It becomes clear later that the authors decide to point out to the model of the second strategy as the significant predictive model. Further explanation is desirable with emphasis on the value added by such approach.
The tables and figures in the manuscript require revision. It is useful to include Figure S1. ‘Model development and validation’ in the manuscript rather than the supplement if the authors decided to continue with the current analysis. In addition, the pooled data was not presented in table 1. Better drafting of Table A1 and A2 would be better if committed to facilitate easier visualization of the comparison (like the format of Table 2).
The data set had some limitation including being old (2001-2012), very wide were changes in practice took place and could be confounder of the outcome with high percentage of missing data. These should be in the limitation as it would affect its external validity especially when this dataset was used for the development of the model. Another limitation is in the eligibility criteria, depending on the ICD-9 diagnosis code for those missing Creatinine level is not accurate and subjective, especially as they received RRT within the first 2 days of ICU admission. The hospital admission creatinine level could help in their identification, especially, if the authors excluded CKD 4-5. AKI severity identification via ICD 9 code is not reliable in my opinion as it depends on the documentation/human factor especially for old records where no validation of the that was done (using the hospital admission creatinine level for example).
The predictor variables selection methodology was not clearly described with significant overlap. Were they significant for the outcome at univariate analysis or they were chosen based on clinical judgment ? Vasopressors use and shock index, used as predictor of shock and possible need of vasopressors, were both included. It was not clear if shock index was used for those without vasopressors or for all patients. Calculation the index while on vasopressors would be misleading, therefore, including both is not advisable. Anion gab is affected by Lactate and urea (result from AKI), therefore, its addition to the model is also not optimal.
The management of missing data needs a closer look. Important variables, used in the models, with very high percentage of missing variable and managed by imputation would affect the results greatly, hindering them undependable. Albumin >51% and Lactate > 34% missing for the dataset used for training and for validation afterwards of the models is not optimal.
ICU diagnosis and other factors, such as infection, use of nephrotoxic medications and the use of IV contrast, are important predictor of AKI and mortality, they were not reported. Especially the models were not designed only on laboratory data, but included more variables
Since both datasets were eventually pooled in the second strategy, their variables results were not reported in the table 1, only the 2 datasets separated.
Were the developed models from the 2nstrategies the same? or were they different? If they were different models, as I concluded given that the top 10 variables contributing the outcome are different moreover those in common have significantly variable weight of prediction. Were the models resulted from both strategies tested using the other testing techniques? Prior to external validation. The exact description of the model was not clear.
The use of strong statements despite the methodology was used in the conclusion and discussion. We disagree with the statement of (excellent performance of ML) while the accuracy was only modest, and (high negative predictive value) while they were less than 80%. We request the authors to revise the conclusion and statements regarding the performance and accuracy. Especially, they have pointed to the potential application of guiding therapy and deciding prognosis.
The manuscript require further work from the authors to be sound and focused with clear flow and outcome.
Author Response
Response to Referee: 3
Comments to the Author
Thank you for the opportunity to review the manuscript titles “Predicting Mortality Using Machine Learning Algorithms in Patients Who Require Renal Replacement Therapy in the Critical Care Unit”. The authors attempted to study the value of machine learning models in predicting mortality in patients receiving renal replacement therapy in the Critical care unit for acute renal injury.
The commonly known scores have been shown to be less accurate in patients requiring RRT in the ICU, therefore the authors have elected to pursue the study of ML utilization for prognostic model development for this purpose. However, we find the methodology was not optimal enough to answer the question. The comparator scores were known to have fair performance, missing data management and methodology of model variables’ selection along with development and testing strategies are some of many to be commented on, we will elaborate further below.
Comments
- The use of two strategies for models’ development using two highly variable dataset did not add value to the methodology. In the first strategy, it is not clear why the model development used the older, single-center, and smaller dataset while testing was on the larger multi-center eICU dataset. In addition, we believe reporting the results of both strategies did not contribute to the aim on the study rather added confusion on my part as a reader. Therefore, despite utilizing the same variables, the two versions of every model of the 4 ML developed by the two strategies were different regarding the ranking of the top 10 predictors along with their effect on outcome (Figure S6). It becomes clear later that the authors decide to point out to the model of the second strategy as the significant predictive model. Further explanation is desirable with emphasis on the value added by such approach.
Response:
We thank the reviewer for allowing us to explain more. Why we chose the MIMIC dataset as training dataset was MIMIC dataset is a widely-used, large, open database. Besides, it is from a tertiary medical center, which means the severity is higher and quality of care is better than general hospitals and the medical records should be more precise. So, the goal of our first strategy is to prove if our models could work in an external validation dataset. The main reason for our secondary strategy is prove if our models could work in a more heterogenous dataset so we mixed eICU and MIMIC datasets to test our models. (line 167 page 8)
- The tables and figures in the manuscript require revision. It is useful to include Figure S1. ‘Model development and validation’ in the manuscript rather than the supplement if the authors decided to continue with the current analysis. In addition, the pooled data was not presented in table 1. Better drafting of Table A1 and A2 would be better if committed to facilitate easier visualization of the comparison (like the format of Table 2).
Response:
We thank the reviewer’s valuable comment. We revised it. (moved previous Figure S1 to Figure 1, moved previous Table A1 and A2 to Supplement Table S3 and S4, New Table 1 shows comparisons between the training and testing datasets of pooled dataset)
- The data set had some limitation including being old (2001-2012), very wide were changes in practice took place and could be confounder of the outcome with high percentage of missing data. These should be in the limitation as it would affect its external validity especially when this dataset was used for the development of the model. Another limitation is in the eligibility criteria, depending on the ICD-9 diagnosis code for those missing Creatinine level is not accurate and subjective, especially as they received RRT within the first 2 days of ICU admission. The hospital admission creatinine level could help in their identification, especially, if the authors excluded CKD 4-5. AKI severity identification via ICD 9 code is not reliable in my opinion as it depends on the documentation/human factor especially for old records where no validation of the that was done (using the hospital admission creatinine level for example).
Response:
We agree with the reviewer that MIMIC dataset is old and practice patterns may have changed, and has missing data, thus we have added these to limitations. (line 317 page 15) Given the limitations of a clinical dataset, we tried to be as broad in our definition of AKI as possible, while minimizing bias. Thus patients were included when they didn’t have at least 2 creatinine data but have AKI as diagnosis or by change in Cr. This dataset is only limited to labs when patient is in the ICU, so it is possible that we may have missed AKI by Cr values if they were admitted to the hospital in non-ICU setting prior to their ICU admission.(line 318 page 15) Since the AKI severity is the worst (stage 3 by KDIGO) when AKI patients require RRT, we included patients who received RRT and did not have CKD4/5 at baseline. We have added this to limitations. (Kellum, John A., and Norbert Lameire. "Diagnosis, evaluation, and management of acute kidney injury: a KDIGO summary (Part 1)." Critical care 17.1 (2013): 1-15.)
- The predictor variables selection methodology was not clearly described with significant overlap. Were they significant for the outcome at univariate analysis or they were chosen based on clinical judgement? Vasopressors use and shock index, used as predictor of shock and possible need of vasopressors, were both included. It was not clear if shock index was used for those without vasopressors or for all patients. Calculation the index while on vasopressors would be misleading, therefore, including both is not advisable. Anion gab is affected by Lactate and urea (result from AKI), therefore, its addition to the model is also not optimal.
Response:
We thank the reviewer’s precious comment. All variables were chosen based on clinical judgement. We re-developed out models after removing vasopressors variable and variables > 30% missing data. Since Anion gap and urea are important factors for AKI patients with dialysis, we kept anion gap in our model.
- The management of missing data needs a closer look. Important variables, used in the models, with very high percentage of missing variable and managed by imputation would affect the results greatly, hindering them undependable. Albumin >51% and Lactate > 34% missing for the dataset used for training and for validation afterwards of the models is not optimal.
Response:
We thank the reviewer’s valuable comment. We re-developed our models after removing variables > 30% missing data.
- ICU diagnosis and other factors, such as infection, use of nephrotoxic medications and the use of IV contrast, are important predictor of AKI and mortality, they were not reported. Especially the models were not designed only on laboratory data, but included more variables
Response:
We thank the reviewer for allowing us to explain more. Those factors are important for AKI.. We believe that although these variables are important predictors for development of AKI, they are likely not directly important in predicting mortality, especially in severe AKI requiring RRT which is usually due to multi-organ failure in the ICU setting. The other clinical, lab and hemodynamic variables in our model will be reflective of disease severity and overall clinical condition and are more important in predicting mortality regardless of etiology of AKI.
- Since both datasets were eventually pooled in the second strategy, their variables results were not reported in the table 1, only the 2 datasets separated.
Response:
We thank the reviewer’s valuable comment. New Table 1 shows comparisons between the training and testing datasets of pooled dataset.
- Were the developed models from the 2nstrategies the same? or were they different? If they were different models, as I concluded given that the top 10 variables contributing the outcome are different moreover those in common have significantly variable weight of prediction. Were the models resulted from both strategies tested using the other testing techniques? Prior to external validation. The exact description of the model was not clear.
Response:
We thank the reviewer for allowing us to explain more. The models for pooled data were the same machine learning algorithms, but were trained using pooled data. Why we used pooled data to train models was to develop generalized models. The weight of variables in different models should be different because those are generated from different machine learning algorithms. We didn’t test our results using the other testing techniques.
- The use of strong statements despite the methodology was used in the conclusion and discussion. We disagree with the statement of (excellent performance of ML) while the accuracy was only modest, and (high negative predictive value) while they were less than 80%. We request the authors to revise the conclusion and statements regarding the performance and accuracy. Especially, they have pointed to the potential application of guiding therapy and deciding prognosis.
Response:
We thank the reviewer for allowing us to explain more. We have softened our language to say that our model had reasonable performance or performance was reasonable high (instead of excellent). (line 253 page 12, line 274 page 13)

Round 2
Reviewer 3 Report
In general:
The authors reaction to the reviewers response resulted in changes in the statistical analysis and more focused and relevant stream of the manuscript result, tables and outcome.
response to comment 1: The goals of the 2 strategies for models should be clarified, which is the primary method for model development, and which used for validation. It not clarified in the methodology section.
response to comment 3:
The authors took the comments in consideration and have revised the limitation section. Inclusion criteria are revised to minimize uncertainty but a new limitation is added which is that of selection bias imposed by the exclusion of missing data on admission of potentially relevant patients. Such data is important and should be mentioned in the limitations as it is in the arena of the limited availability of clinical and centers' setting data could represent potential confounders and result in selection bias related inaccurate outcome.
I failed to find the reference eluded to by the authors in (line 317) regarding their editing.
Tthe inclusion methodology of AKI needs to be rewritten in clearer form as their response is not clarified well in the inclusion crtieria section (line 101 forward, page 5) as well as, (AKI defined by creatinine change level) timelines are not clear (? from hospital admission to RRT or to ICU admission) as they described in (line 104, page 5) that those missing the previous data but 2 creatinine level in the icu were available, was utilized to calculate the change in level. in addition to the above, the response provided by the authors to comment 3 should be better reflected in the manuscript.
response to comment 4:
It would be of interest of these variables included on the ML were significantly associated with the desired outcome utilizing uni- and multiple logistic regression in the first place or no. The results of that should be discussed accordingly. Depending on the clinical judgment solely is not sufficient and impose significant chance of bias and potential of overlooking other significant variables. The authors may choose to include the results in supplementary file, however, the methodology should be discussed in the limitation and how that was managed.
In addition, the decision to pursue the exclusion of the use of vasopressors, without statistical evidence at least of its insignificance, while keeping the shock index and mean arterial pressure (MAP), which could be normal in case vasopressors administration, is not wise and should be defended well especially if all the predictors included in the model were selected based on authors judgment to start with. Especially Vasopressors were retained as important predictor in the precious version of the manuscript (Figure 3 from V1)
for comment 5, In addition to the mentioned factors, the use of vasopressors is a significant one and is included in the model, it would be interesting to check weather the association of modality choice of RRT on mortality is independent of the vasopressors use.
The authors indicated a difference in the modalities used for RRT and their frequencies of their use among the groups stratified according the outcome. Was the factors affecting the likely hood of HD vs CRRT studied? Would the use of vasopressors be an indictors and was the modality chosen significant for mortality?
comment 6 response:
Sepsis in the ICU is associated with higher mortality is a known fact. If the authors believe sepsis related AKI association with mortality is only directly related through the use of RRT, then clear literature review is needed to defend it. In general, the clinical cause of AKI and association with higher mortality was also pointed to by the authors in their response to comment 1 which also touch on the diagnosis and etiology of AKI and their effect on the outcome measured, mortality.
Abstract needs to be revised to reflect the revisions made to the manuscript. For example; line43 reads (we developed a series of machine learning models to improve prediction accuracy of mortality and compared their performance to SOFA ...etc) should be re-edited.
the conclusion language was soften , according to the comment 9 response, but still strong in the abstract and aim section in the introduction.
Does figure 2 needs update following the revision of methodology?
Thank you for the opportunity to review the manuscript.
